# Biological Activity and In Silico Study of 3-Modified Derivatives of Betulin and Betulinic Aldehyde

**DOI:** 10.3390/ijms20061372

**Published:** 2019-03-19

**Authors:** Ewa Bębenek, Elwira Chrobak, Krzysztof Marciniec, Monika Kadela-Tomanek, Justyna Trynda, Joanna Wietrzyk, Stanisław Boryczka

**Affiliations:** 1Medical University of Silesia in Katowice, School of Pharmacy with the Division of Laboratory Medicine in Sosnowiec, Department of Organic Chemistry, 4 Jagiellońska Str., 41-200 Sosnowiec, Poland; echrobak@sum.edu.pl (E.C.); kmarciniec@sum.edu.pl (K.M.); mkadela@sum.edu.pl (M.K.-T.); boryczka@sum.edu.pl (S.B.); 2Wroclaw University of Environmental and Life Science, Department of Experimental Biology, 27b Norwida Str., 50-375 Wrocław, Poland; justyna.trynda@upwr.edu.pl; 3Polish Academy of Sciences, Ludwik Hirszfeld Institute of Immunology and Experimental Therapy, Department of Experimental Oncology, 12 Rudolfa Weigla Str., 53-114 Wrocław, Poland; wietrzyk@iitd.pan.wroc.pl

**Keywords:** betulin, betulinic aldehyde, antiproliferative activity, ADME, molecular docking

## Abstract

A series of 3-substituted derivatives of betulin and betulinic aldehyde were synthesized as promising anticancer agents. The newly triterpenes were tested against five human cancer cell lines like biphenotypic B myelomonocytic leukaemia (MV-4-11), adenocarcinoma (A549), prostate (Du-145), melanoma (Hs294T), breast adenocarcinoma (MCF-7) and normal human mammary gland (MCF-10A). The compound **9** showed towards Du-145, MCF-7 and Hs294T cells significant antiproliferative activity with IC_50_ ranging from 7.3 to 10.6 μM. The evaluation of ADME properties of all compounds also includes their pharmacokinetic profile. The calculated TPSA values for synthetized derivatives are in the range between 43.38 Å^2^ and 55.77 Å^2^ suggesting high oral bioavailability. The molecular docking calculations showed that triterpene **9** fits the active site of the serine/threonine protein kinase Akt.

## 1. Introduction

Presently, more than 120 drugs of plant origin are used for the treatment or amelioration of various diseases. Many promising candidates for medicines derived from natural sources have some undesirable properties, including poor water solubility and low thermal stability. However, these substances can be used as lead compounds for the preparation of new drugs [1].

The large class of natural compounds are triterpenes, the secondary plant metabolites, represented by betulin **1**. The outer bark layer of the birch tree contain as major compound betulin **1** (Figure 1) (12–30% of dry weight) and trace amounts of other triterpenes such as allobetulin, betulinic acid, betulinic aldehyde and lupeol [2]. The biological properties reported for betulin **1** and its derivatives include anticancer [3,4], antiviral [5,6], antibacterial [7,8] and anti-inflammatory activity [9,10,11].

The structural diversity of triterpenes are obtained generally through chemical modification of their functional groups at the C-3, C-28, C-20 and C-17 positions. The promising semisynthetic derivatives of triterpenes as anticancer agents were obtained by introduction various functional groups at the C-3 position. An example of such substances are 3-methyl- and 3-ethyl-phthalates of betulinic acid showed significant cytotoxicity compared to betulinic acid towards six tested cancer cell lines. The most active compound was 3-methyl-phthalate exhibited cytotoxic activity towards cancer cells from different histogenetic origins like epithelial tumours (PC-3, HT-29), neuroectodermal tumours (SK-MEL2) and mesenchymal tumours (K562, K562-tax, CEM) [12]. Received by [13] 3-glutaryl-, 3-succinyl- and 3-acetylbetulinic acids exhibited higher activity than the betulinic acid (IC_50_ = 8.4 µg/mL) against the lung A549 cell line with IC_50_ ranging from 6.4 to 7.4 µg/mL.

Moreover, it was observed that the derivatives of betulinic acid bearing a shorter alkyl chain on the acyl group at the C-3 position were more toxic on the A549 cells [13]. Cytotoxic data of the 3-modified derivatives of dihydrobetulinic acid indicates that presence of the 4-fluorophenyl-hydrazono moiety at the C-3 position led to compound with significant cytotoxic activity towards cells of leukaemia (MOLT-4), ovary (PA-1) and colon (HT-29) cancer. Additionally, it appeared, that derivative containing the 4-fluorophenyl-hydrazono group at the C-3 position had better water solubility (124.7 μM) as compared to betulinic acid (<1 μM) [14]. The introduction the triazolyl moiety at the C-3 position of betulin **1** and 2′-methylimidazolyl moiety at the C-3 position of betulinic acid produce a novel derivatives of triterpenes with potent anticancer activity. The cytotoxic activity of 3-substituted tiazole and imidazole derivatives was evaluated towards different tumours type such as prostate adenocarcinoma (PC-3), leukaemia (Jurkat), cervical adenocarcinoma (HeLa), hepatocellular carcinoma (HepG2) and colon adenocarcinoma (HT-29). These compounds showed a better cytotoxic profile than the betulinic acid (IC_50_ = 12.8–36.4 μM) with IC_50_ values ranging from 2.2 to 12.2 μM in the tested cancer cell lines [15].

The addition of polar group at the C-3 position of betulinic acid such as α-l-rhamnopyranose moiety increase the water solubility and hence the antiproliferative activity against two applied human cancer cell lines A549 (lung carcinoma) and DLD-1 (colon adenocarcinoma) [16]. Cui et al. synthesized a new 3-substituted derivatives of betulinic acid by introducing nitrogen-containing heterocycles connected to the triterpene skeleton with an amide linkage. The all 3-amide derivatives of betulinic acid were tested for their in vitro antiproliferative activity towards eight human cell lines like prostate cancer (Du-145, PC-3), colon carcinoma (HT-29, HCT-116), breast cancer (T47D, MCF-7, MDA-MB-231) and multidrug-resistant breast cancer (MCF-7/ADR). Especially, derivative of betulinic acid containing the 4-piperidinecarboxamide group at the C-3 position had the significant cytotoxic activity (IC_50_ = 0.33 µM) towards MCF-7/ADR cells, about 117-fold more potent than the betulinic acid (IC_50_ = 38.5 µM). It was demonstrated, that this compound induced apoptosis on the MCF-7 and MCF-7/ADR cell lines and exhibited potent antimetastatic activity on the MDA-MB-231 cells [17].

Based on the above information and our previous studies we synthesized the newly 3-substituted derivatives of betulin and betulinic aldehyde with alkyl, alkenyl and alkynyl chain on the acyl group [18,19,20,21]. The antiproliferative activity in vitro was evaluated towards five human cancer cell lines like biphenotypic B myelomonocytic leukaemia (MV-4-11), adenocarcinoma (A549), prostate (Du-145), melanoma (Hs294T), breast adenocarcinoma (MCF-7) and normal human mammary gland (MCF-10A) by the MTT and the SRB methods. The predicted ADME properties were used to determinate of bioavailability of newly 3-substituted triterpenes. Additionally, the molecular docking studies were conducted to identify the binding mechanism of obtained compounds with the serine/threonine kinase Akt.

## 2. Results and Discussion

### 2.1. Chemistry

Firstly, the naturally occurring betulin **1** was used as starting material to the synthesis of 3-acetylbetulin **2** and 3-acetylbetulinic aldehyde **3** (Figure 1). The synthesis of compounds **2**–**3** was carried out on the basis of the procedure described by Liu et al. [22].

As shown in Scheme 1, the monoprotection of betulin **1** by pyridinium p-toluenesulfonate (PPTS) and dihydropyran (DHP) in dichloromethane gave the tetrahydropyranyl ether **4** in 57% yield [23,24]. The series of 3-substituted betulin derivatives **5**–**14** was obtained in the two-step synthesis involving the esterification reaction of the secondary hydroxyl group at the C-3 position followed by the deprotection of the THP group with PPTS in ethanol. 

The resulting esters **5**–**14** were oxidized by pyridinium chlorochromate (PCC) in dry dichloromethane to the derivatives of betulinic aldehyde **15**–**24** in 61–84% yields. The chemical structures of all obtained triterpenes **5**–**24** were confirmed by ^1^H NMR, ^13^C NMR, IR and HR-MS techniques (Appendix A).

### 2.2. Antiproliferative Activity

In the first stage of the in vitro antiproliferative activity research the screening test of the compounds **5–24** towards the human biphenotypic B myelomonocytic leukaemia (MV-4-11) cell line was performed. Betulin **1**, 3-acetylbetulin **2**, 3-acetylbetulinic aldehyde **3** and cisplatin were applied as the reference compounds. The results of cytotoxicity of the derivatives **5**–**24** towards the MV-4-11 cells are reported in Table 1 as IC_50_ (µM). It has been observed, that presence of an alkenyl and alkynyl moiety at the C-3 position of betulin **1** molecule moderately increased cytotoxicity towards MV-4-11 cells. The rank order of the anticancer activity against MV-4-11 is as follows: **6** > **9** > **16** > **13** > **14** > **10** > **12** > **20** > **24** > **23** > **11** > **19** > **22** > **5** > **7** > **15** > **21** > **17** > **8** = **18**.

The compound **6** having 2-propenoyl group showed the highest anticancer activity with IC_50_ value of 4.2 μM. The 3-alkynyl derivatives **9** and **13–14** exhibited more potent cytotoxicity than 3-acetylbetulin **2** with IC_50_ values ranging from 4.5 to 8.9 μM. The remaining compounds showed moderate activity towards MV-4-11 with IC_50_ value above 10 μM. It should be noted, that the presence of the formyl group at the C-17 position of compounds **15**–**24**, resulted in decrease of activity towards MV-4-11 cells compared to their 3-substituted analogues **5**–**14**. In the case of derivatives of betulinic aldehyde, only compound **16** exhibited significant activity against MV-4-11 cells, 6-fold more potent than the 3-acetylbetulinic aldehyde **3**.

Among the tested compounds only those with level of the anticancer activity which was close or higher to betulin **1**, were selected to the second stage of the study. The compounds **6**, **9**, **13**–**14**, **16**, betulin **1** and cisplatin were evaluated in vitro for antiproliferative activity against human cell lines like adenocarcinoma (A549), prostate (Du-145), melanoma (Hs294T), breast adenocarcinoma (MCF-7) and normal mammary gland (MCF-10A). The results of antiproliferative activity of the triterpenes **6**, **9**, **13**–**14**, **16** against five applied cell lines (A549, Du-145, Hs294T, MCF-7 and MCF-10A) are presented in Table 2.

Our studies showed moderate activity of derivatives **6**, **13**–**14**, **16** against the tested cell lines. The most active compound was derivative **9** for which the IC_50_ values are in the range 7.3–83.5 μM. The introduction of the 2-butynoyl group at the C-3 position in compound **9** resulted in an increase of antiproliferative activity in Du-145, Hs294T and MCF-7 cell lines when compared with betulin **1**. Figure 2 present the Selectivity Index (SI) calculated for compounds **6**, **9**, **16** and betulin **1** according to the formula: SI = IC_50_ for normal mammary gland/IC_50_ on corresponding tumour cell line. Additionally, triterpene **9** was more selective towards Du-145 (SI= 9.82), Hs294T (SI = 6.76) and MCF-7 (SI = 6.89) cells than the reference compound **1**. The obtained results exhibited that carbon-carbon triple bond of carboxyl moiety at the C-3 position is favourable for the antiproliferative activity.

### 2.3. In Silico Study

Lipinski’s “Rule of five” and Veber’s criteria are used to assess the similarity of a compound to a drug substance [25,26,27]. This is related to the determination of such properties as blood (plasma)-brain partition coefficient permeant (logBBB), lipophilicity (logP), number of hydrogen bond acceptors and donors (HBA and HBD) and topological polar surface area (TPSA). The ADME parameters of the newly synthesized triterpenes are detailed in Table 3. 

The derivatives **5**–**24** demonstrate the octanol water partition coefficient (logP) in the range 7.67–9.72, which may be evidence of low ability of compounds to cross the cell membranes. As can be seen in the Table 3, the HBA and HBD values of the 3-substituted derivatives are within the preferred range (HBA ≤ 10 and HBD ≤ 5). The number of hydrogen bond acceptors (HBA) and hydrogen bond donors (HBD) of the compounds **5**–**24** are in the range 0–4.

The TPSA values for all compounds are found in the range between 43.38 Å^2^ and 55.77 Å^2^, which indicates their high oral bioavailability. In addition, low values TPSA of less than 60 Å^2^ shows good permeability through the Blood-Brain Barrier (BBB). The all triterpenes **5**–**24** exhibit the logBB > −1, which suggest their good transport to the brain. The results of in silico study shows, that these compounds can be considered as promising candidates for the treatment of the central nervous system (CNS).

Triterpenoids exhibit antiproliferative activity against various cancer cells in vivo, though their precise molecular target it is often difficult to determine. Molecular docking studies showed significant interaction of this class compounds with important anticancer targets such as topoisomerases I and IIα, PPARγ, EGFR, DHFR, VEGFR, NF-κβ, HER-2/neu, hCA-IX, CDK6 and LOX [28,29,30,31,32,33].

Molecular docking study of compound **9** was performed based on predictions of biological activity obtained with use of PASS computer program (Prediction of Activity Spectra for Substances) [34].

The predicted biological activity is determined based on the probable activity (Pa) and probable inactivity (Pi). The values of Pa > 0.7 indicate a high probability of a specific activity, which gives a good chance to confirm it in biological studies. The PASS prediction showed anticancer activity of triterpene **9** against lung (Pa > 0.81), melanoma (Pa > 0.80) and breast (Pa > 0.74) cancers in which there is an increased activity of the Akt protein. Additionally, the results obtained for compound **9** from the PASS program confirmed the studies of antiproliferative activity against Hs294T and MCF-7 cells.

The serine/threonine kinase Akt is the major signal transducer in the phosphatidylinositol 3-kinase (PI 3-K) pathway. Akt plays an important role in the cellular processes related with cancer like cell growth, proliferation, metabolism and angiogenesis. Increased expression of the Akt protein was observed in various human cancers, for example, breast, prostate, lung, ovary and brain cancer [35,36,37].

Molecular docking techniques were applied to investigate the binding mechanism for the series of triterpene derivatives with Akt. In our studies we retrieved the crystal structure of Akt1 from the RCSB Protein Data Bank (PDB ID: 3QKK). Docking results obtained from the GOLD program for Akt1 showed a higher degree of fit defined in the arbitrary units of the program for triterpene **9** (docking score of 57.55) compared to reference betulin **1** (docking score of 55.44). However, the alkynyl derivative **9** took a different place in the binding pocket than parent betulin **1**. The pentacyclic moiety of betulin **1** was located deep in hydrophobic cavity of CAT domain of Akt formed by Phe295/Phe299/Arg331. Significant difference in location compound **9** in CAT domain was observed.

The chemistry of the active site cavity in enzyme shows that bounded 2-butynoyl group have deeply penetrated the strong hydrophobic matrix of these proteins. Figure 3 presents the geometry of the Akt1 binding cavities, with betulin **1** and compound **9** buried in the protein’s hydrophobic environment.

Molecular shape and hydrophobic behaviour of triterpene **9** seems to be a major factor responsible for its potent inhibitory activity (Figure 4).

Favourable hydrophobic interactions were detected between Phe18 and methyl group of the triterpene skeleton at C-8 position at the distance 4.83 Å (Figure 5).

On the opposite side, Phe299 was favourably supported by interaction with the C-4 of the 2-butynoyl group at a distance 4.44 Å. Leu38 also exhibited hydrophobic interaction with ethenyl carbon at C-29 position, as well as Arg331 with methyl group at C-4 position at a distance 4.54 and 5.16 Å, respectively.

## 3. Materials and Methods

### 3.1. General Techniques

Organic solvents were procured commercially and used after purification. Melting points (m.p.) of the obtained triterpenes were determined in open capillary tubes on an Electrothermal IA 9300 melting point apparatus and are uncorrected. Optical rotations were measured with an Atago Sac-I polarimeter (Atago, Tokyo, Japan). The NMR (^1^H and ^13^C) spectra were recorded on a Bruker Avance (Bruker, Billerica, MA, USA) III 600 spectrometer in deuterated- chloroform (δ given in ppm, *J* in hertz). The HR-MS of all compounds were recorded on a Bruker Impact II instrument (Bruker, Billerica, MA, USA). The IR spectra were recorded on a Shimadzu IRAffinity-1 FTIR spectrophotometer (Shimadzu, Kyoto, Japan). The progress of the reactions was monitored by thin layer chromatography (TLC) using silica gel 60 254F plates (Merck, Darmstadt, Germany). The spots were detected by spraying with a solution of 5% sulfuric (VI) acid and heating to 120 °C. Silica gel 60, <63 μm (Merck) were used for column chromatography. A mixture of CH_2_Cl_2_-EtOH (40:1 and 60:1, *v*/*v*) was applied as the mobile phase.

### 3.2. Synthesis of 3-Acetylbetulin ***2*** and 3-Acetylbetulinic Aldehyde ***3***

Synthesis of 3-acetylbetulin **2** and 3-acetylbetulinic aldehyde **3** were carried out based on the procedures described in the literature [22].

### 3.3. Synthesis of 28-Tetrahydropyranylbetulin ***4***

To a mixture of betulin **1** (0.44 g, 1 mmol) in dichloromethane (15 mL) was added DHP (0.10 mL, 1.12 mmol) and PPTS (0.03 g, 0.12 mmol). The mixture was stirred at room temperature for 3 day in an inert gas atmosphere. Next, the resulting mixture was quenched by addition of saturated sodium bicarbonate (5 mL). The organic layer was washed with brine (5 mL), dried with anhydrous sodium sulphate and concentrated under reduced pressure. The diastereomeric mixture of ether **4** was purified by column chromatography (CH_2_Cl_2_-EtOH 40:1, *v*/*v*) to give compound **4** with 57% yield [23,24].

### 3.4. General Procedure for the Synthesis of Betulin Derivatives ***5**–**9***

To a mixture of ether **4** (0.26 g, 0.5 mmol) and 0.55 mmol of corresponding carboxylic acid in dichloromethane (2.5 mL) was added a solution of DDC (0.11 g, 0.56 mmol) and DMAP (0.005 g, 0.04 mmol) in dichloromethane (0.5 mL) under argon atmosphere at −10 °C temperature. The reaction was continued to stir at −10 °C temperature for 5h and then was raised to room temperature and stirred overnight. The solvent was removed under reduced pressure. The obtained residue was dissolved in ethanol (17 mL) and added PPTS (0.21 g, 0.87 mmml). The reaction mixture was agitated at room temperature for one week. Next, the resulting mixture was quenched by addition of saturated bicarbonate (5 mL) and extracted dichloromethane (3 × 10 mL). The organic layer was washed with water (4 × 10 mL), dried with anhydrous sodium sulphate and concentrated under reduced pressure. The crude residue was purified by column chromatography (CH_2_Cl_2_-EtOH 60:1, *v*/*v*) to afford compounds **5**–**9** with 56–85% yields.

3-Propanoylbetulin (**5**) Yield 56%; mp 252–254 °C; [α]_D_^20^ +1.8 (c 1, CHCl_3_); R_f_ 0.31 (dichloromethane/ethanol, 60:1, *v*/*v*); IR (KBr) ν_max_ 3415, 2969, 1733, 1465, 1266 cm^−1^; ^1^H NMR (600 MHz, CDCl_3_): δ 4.71 (1H, s, H-29), 4.61 (1H, s, H-29), 4.50 (1H, m, H-3), 3.81 (1H, d, *J* = 10.8 Hz, H-28), 3.35 (1H, d, *J* = 10.8 Hz, H-28), 2.40 (1H, m, H-19), 2.34 (2H, q, *J* = 7.2 Hz, *CH*_2_CH_3_), 1.69 (3H, s, CH_3_), 1.17 (3H, t, *J* = 7.2 Hz, CH_2_*CH*_3_), 1.06 (3H, s, CH_3_), 1.00 (3H, s, CH_3_), 0.88 (3H, s, CH_3_), 0.87 (3H, s, CH_3_), 0.86 (3H, s, CH_3_); ^13^C NMR (150 MHz, CDCl_3_): δ 173.3, 149.4, 108.7, 79.6, 59.5, 54.4, 49.3, 47.7, 46.8, 46.8, 41.7, 39.9, 37.4, 36.9, 36.3, 36.1, 33.1, 32.9, 28.7, 28.1, 27.1, 26.9, 26.0, 24.2, 22.7, 19.8, 18.0, 17.2, 15.5, 15.1, 14.9, 13.7, 8.3; HRAPCIMS *m*/*z*: 497.4002 C_33_H_53_O_3_ (calcd. 497.3995).

3-(2-Propenoyl)betulin (**6**) Yield 81%; mp 221–223 °C; [α]_D_^20^ +3.3 (c 1, CHCl_3_); R_f_ 0.33 (dichloromethane/ethanol, 60:1, *v*/*v*); IR (KBr) ν_max_ 3429, 2943, 1720, 1457, 1275 cm^−1^; ^1^H NMR (600 MHz, CDCl_3_): δ 6.39 (1H, m, CH=*CH*_2_), 6.13 (1H, m, *CH*=CH_2_), 5.82 (1H, m, CH=*CH*_2_), 4.70 (1H, s, H-29), 4.61 (1H, s, H-29), 4.58 (1H, m, H-3), 3.82 (1H, d, *J* = 10.8 Hz, H-28), 3.35 (1H, d, *J* = 10.8 Hz, H-28), 2.42 (1H, m, H-19), 1.69 (3H, s, CH_3_), 1.06 (3H, s, CH_3_), 1.01 (3H, s, CH_3_), 0.89 (3H, s, CH_3_), 0.88 (3H, s, CH_3_), 0.87 (3H, s, CH_3_); ^13^C NMR (150 MHz, CDCl_3_): δ 166.1, 150.5, 130.0, 129.2, 109.7, 81.1, 60.6, 55.4, 50.3, 48.8, 47.8, 47.7, 42.7, 40.9, 38.4, 38.0, 37.3, 37.1, 34.2, 34.0, 29.7, 29.2, 28.0, 27.0, 25.2, 23.7, 20.9, 20.0, 19.1, 18.2, 16.6, 16.2, 16.0, 14.7; HRAPCIMS *m*/*z*: 495.3836 C_33_H_51_O_3_ (calcd. 495.3838).

3-(3-Cyclopropyl-2-propynoyl)betulin (**7**) Yield 63%; mp 267–269 °C; [α]_D_^20^ +2.5 (c 1, CHCl_3_); R_f_ 0.33 (dichloromethane/ethanol, 60:1, *v*/*v*); IR (KBr) ν_max_ 3530, 2943, 2224, 1691, 1452, 1276 cm^−1^; ^1^H NMR (600 MHz, CDCl_3_): δ 4.70 (1H, s, H-29), 4.61 (1H, s, H-29), 4.58 (1H, m, H-3), 3.81 (1H, d, *J* = 10.8 Hz, H-28), 3.35 (1H, d, *J* = 10.8 Hz, H-28), 2.42 (1H, m, H-19), 1.68 (3H, s, CH_3_), 1.04 (3H, s, CH_3_), 0.93 (3H, s, CH_3_), 0.94–0.99 (5H, m, CH, CH_2_), 0.89 (3H, s, CH_3_), 0.88 (3H, s, CH_3_), 0.87 (3H, s, CH_3_); ^13^C NMR (150 MHz, CDCl_3_): δ 154.5, 151.0, 110.3, 93.2, 83.2, 69.4, 61.1, 55.9, 50.8, 49.3, 48.4, 48.3, 43.2, 41.5, 38.9, 38.5, 37.8, 37.6, 34.7, 34.5, 30.3, 29.7, 28.4, 27.6, 25.7, 24.1, 21.4, 19.6, 18.7, 17.1, 16.7, 16.5, 15.2, 9.7, 1.6, HRAPCIMS *m*/*z*: 533.3985 C_36_H_53_O_3_ (calcd. 533.3995).

3-Phenylpropynoylbetulin (**8**) Yield 83%; mp 217–219 °C; [α]_D_^20^ +4.4 (c 1, CHCl_3_); R_f_ 0.32 (dichloromethane/ethanol, 60:1, *v*/*v*); IR (KBr) ν_max_ 3482, 2940, 2218, 1699, 1452, 1286 cm^−1^; ^1^H NMR (600 MHz, CDCl_3_): δ 7.51–7.29 (5H, m, Ar-H), 4.62 (1H, s, H-29), 4.58 (1H, m, H-3), 4.52 (1H, s, H-29), 3.72 (1H, d, *J* = 10.8 Hz, H-28), 3.26 (1H, d, *J* = 10.8 Hz, H-28), 2.32 (1H, m, H-19), 1.67 (3H, s, CH_3_), 0.98 (3H, s, CH_3_), 0.96 (3H, s, CH_3_), 0.90 (3H, s, CH_3_), 0.89 (3H, s, CH_3_), 0.84 (3H, s, CH_3_); ^13^C NMR (150 MHz, CDCl_3_): δ 154.2, 150.5, 132.9, 130.5, 128.5, 119.9, 109.8, 85.8, 83.2, 81.1, 60.5, 55.4, 50.3, 48.7, 47.8, 47.8, 42.7, 40.9, 38.4, 38.0, 37.3, 37.1, 34.1, 33.9, 29.7, 29.1, 27.9, 27.0, 25.1, 23.6, 20.9, 19.1, 18.2, 16.6, 16.2, 15.9, 14.7; HRAPCIMS *m*/*z*: 585.3925 C_39_H_53_O_3_ (calcd. 585.3995).

3-(2-Butynoyl)betulin (**9**) Yield 85%; mp 233–236 °C; [α]_D_^20^ +1.6 (c 1, CHCl_3_); R_f_ 0.34 (dichloromethane/ethanol, 60:1, *v*/*v*); IR (KBr) ν_max_ 3580, 2946, 2243, 1699, 1457, 1260 cm^−1^; ^1^H NMR (600 MHz, CDCl_3_): δ 4.70 (1H, s, H-29), 4.60 (1H, s, H-29), 4.58 (1H, m, H-3), 3.81 (1H, d, *J* = 10.8 Hz, H-28), 3.34 (1H, d, *J* = 10.8 Hz, H-28), 2.44 (1H, m, H-19), 2.00 (3H, s, C≡C*CH*_3_), 1.70 (3H, s, CH_3_), 1.03 (3H, s, CH_3_), 0.98 (3H, s, CH_3_), 0.89 (3H, s, CH_3_), 0.87 (3H, s, CH_3_), 0.86 (3H, s, CH_3_); ^13^C NMR (150 MHz, CDCl_3_): δ 153.9, 150.5, 109.8, 85.0, 82.8, 72.8, 60.6, 55.4, 50.3, 48.7, 47.8, 42.7, 40.9, 38.4, 37.9, 37.3, 37.0, 34.1, 34.0, 29.7, 29.1, 27.8, 27.0, 25.1, 23.6, 20.8, 19.1, 18.2, 16.5, 16.1, 16.0, 14.7, 3.9, 1.0; HRAPCIMS *m*/*z*: 507.3835 C_34_H_51_O_3_ (calcd. 507.3838).

### 3.5. General Procedure for the Synthesis of Betulin Derivatives ***10**–**14***

To a stirred mixture of ether **4** (0.26 g, 0.05 mmol) in benzene (3 mL) in the presence of pyridine (1.25 mL) at 0–5 °C temperature was added solution of corresponding chloroformate (1.5 mmol) in benzene (2.5 mL). The reaction was continued to stir at 0–5 °C temperature for 4 h. Next, the reaction was raised to room temperature and stirred overnight. The reaction mixture was diluted with 2.5 mL of chloroform and washed with 1 N sulfuric acid and water. After drying with anhydrous sodium sulphate the solution was concentrated under reduced pressure. The obtained residue was dissolved in ethanol (17 mL) and added PPTS (0.21 g, 0.87 mmml). The reaction mixture was agitated at room temperature for one week. Next, the resulting mixture was quenched by addition of saturated bicarbonate (5 mL) and extracted dichloromethane (3 × 10 mL). The organic layer was washed with water (4 × 10 mL), dried with anhydrous sodium sulphate and concentrated under reduced pressure. The crude residue was purified by column chromatography (CH_2_Cl_2_-EtOH 60:1, *v*/*v*) to afford compounds **10**–**14** with 67–75% yields.

3-Ethoxycarbonylbetulin (**10**) Yield 67%; mp 239–242 °C; [α]_D_^20^ +3.5 (c 1, CHCl_3_); R_f_ 0.29 (dichloromethane/ethanol, 60:1, *v*/*v*); IR (KBr) ν_max_ 3497, 2943, 1737, 1466, 1269 cm^−1^; ^1^H NMR (600 MHz, CDCl_3_): δ 4.70 (1H, s, H-29), 4.61 (1H, s, H-29), 4.33 (1H, m, H-3), 4.21 (2H, q, *J* = 7.2 Hz, O*CH*_2_), 3.81 (1H, d, *J* = 10.8 Hz, H-28), 3.35 (1H, d, *J* = 10.8 Hz, H-28), 2.41 (1H, m, H-19), 1.69 (3H, s, CH_3_), 1.31 (3H, t, *J* = 7.2 Hz, CH_2_*CH*_3_), 1.05 (3H, s, CH_3_), 1.00 (3H, s, CH_3_), 0.94 (3H, s, CH_3_), 0.88 (3H, s, CH_3_), 0.87 (s, CH_3_, 3H); ^13^C NMR (150 MHz, CDCl_3_): δ 155.3, 150.5, 109.7, 85.1, 63.7, 60.6, 55.4, 50.3, 48.7, 47.8, 47.7, 42.7, 40.9, 38.4, 38.0, 37.3, 37.1, 34.1, 34.0, 29.7, 29.1, 27.8, 27.0, 25.2, 23.7, 20.1, 19.1, 18.1, 16.4, 16.1, 16.0, 14.7, 14.3; HRAPCIMS *m*/*z*: 513.3940 C_33_H_53_O_4_ (calcd. 513.3944).

3-Propoxycarbonylbetulin (**11**) Yield 69%; mp 250–253 °C; [α]_D_^20^ +4.5 (c 1, CHCl_3_); R_f_ 0.37 (dichloromethane/ethanol, 60:1, *v*/*v*); IR (KBr) ν_max_ 3497, 2953, 1736, 1456, 1257 cm^−1^; ^1^H NMR (600 MHz, CDCl_3_): δ 4.70 (1H, s, H-29), 4.60 (1H, s, H-29), 4.32 (1H, m, H-3), 4.11 (2H, t, *J* = 6.6 Hz, O*CH*_2_), 3.81 (1H, d, *J* = 10.8 Hz, H-28), 3.35 (1H, d, *J* = 10.8 Hz, H-28), 2.41 (1H, m, H-19), 1.73 (2H, m, *CH*_2_CH_3_), 1.67 (3H, s, CH_3_), 1.08 (3H, s, CH_3_), 1.07 (3H, s, CH_3_), 1.05 (3H, t, *J* = 7.2 Hz, CH_2_*CH*_3_), 1.01 (3H, s, CH_3_), 0.97 (3H, s, CH_3_), 0.87 (s, CH_3_, 3H); ^13^C NMR (150 MHz, CDCl_3_): δ 155.5, 150.5, 109.7, 85.1, 69.3, 60.5, 55.4, 50.3, 48.7, 47.8, 48.7, 42.7, 40.9, 38.4, 38.1, 37.3, 37.1, 34.1, 33.9, 29.7, 29.2, 27.9, 27.0, 25.2, 23.7, 22.1, 19.1, 18.1, 16.4, 16.1, 16.0, 14.7, 10.2; HRAPCIMS *m*/*z*: 527.4089 C_34_H_55_O_4_ (calcd. 527.4100).

3-Allyloxycarbonylbetulin (**12**) Yield 69%; mp 220–222 °C; [α]_D_^20^ +1.7 (c 1, CHCl_3_); R_f_ 0.29 (dichloromethane/ethanol, 60:1, *v*/*v*); IR (KBr) ν_max_ 3501, 2942, 1741, 1456, 1253 cm^−1^; ^1^H NMR (600 MHz, CDCl_3_): δ 5.97 (1H, m, *CH*=CH_2_), 5.38 (1H, m, CH=*CH*_2_), 5.28 (1H, m, CH=*CH*_2_), 4.71 (1H, s, H-29), 4.63 (2H, m, O*CH*_2_), 4.61 (1H, s, H-29), 4.34 (1H, m, H-3), 3.82 (1H, d, *J* = 10.8 Hz, H-28), 3.35 (1H, d, *J* = 10.8 Hz, H-28), 2.41 (1H, m, H-19), 1.67 (3H, s, CH_3_), 1.05 (3H, s, CH_3_), 1.01 (3H, s, CH_3_), 0.94 (3H, s, CH_3_), 0.88 (3H, s, CH_3_), 0.87 (3H, s, CH_3_);^13^C NMR (150 MHz, CDCl_3_): δ 155.1, 150.5, 131.9, 118.7, 109.7, 85.5, 68.2, 60.5, 55.4, 50.3, 48.7, 47.8, 47.7, 42.7, 40.9, 38.4, 38.1, 37.3, 37.1, 34.1, 34.0, 29.7, 29.2, 27.9, 27.0, 25.2, 23.7, 20.9, 19.1, 18.1, 16.4, 16.1, 16.0, 14.7; HRAPCIMS *m*/*z*: 525.3937 C_34_H_53_O_4_ (calcd. 525.3944).

3-Propargyloxycarbonylbetulin (**13**) Yield 74%; mp 181–182 °C; [α]_D_^20^ +1.8 (c 1, CHCl_3_); R_f_ 0.33 (dichloromethane/ethanol, 60:1, *v*/*v*); IR (KBr) ν_max_ 3434, 3309, 2945, 2131, 1747, 1456, 1254 cm^−1^; ^1^H NMR (600 MHz, CDCl_3_): δ 4.65 (2H, d, *J* = 2.4 Hz, O*CH*_2_), 4.61 (1H, s, H-29), 4.51 (1H, s, H-29), 4.26 (1H, m, H-3), 3.72 (1H, d, *J* = 10.8 Hz, H-28), 3.26 (1H, d, *J* = 10.8 Hz, H-28), 2.45 (1H, t, *J*= 2.4 Hz, C≡*CH*), 2.31 (1H, m, H-19), 1.67 (3H, s, CH_3_), 0.96 (3H, s, CH_3_), 0.94 (3H, s, CH_3_), 0.89 (3H, s, CH_3_), 0.85 (3H, s, CH_3_), 0.78 (3H, s, CH_3_); ^13^C NMR (150 MHz, CDCl_3_): δ 154.6, 150.5, 109.8, 86.3, 75.4, 60.5, 55.4, 54.9, 50.3, 48.7, 47.8, 42.7, 40.9, 38.3, 38.1, 37.3, 37.0, 34.1, 33.9, 29.7, 29.1, 27.8, 27.0, 25.1, 23.6, 20.8, 19.1, 18.1, 16.4, 16.1, 16.0, 14.7; HRAPCIMS *m*/*z*: 523.3788 C_34_H_51_O_4_ (calcd. 523.3787).

3-(3-Butynyloxycarbonyl)betulin (**14**) Yield 75%; mp 184–186 °C; [α]_D_^20^ +1.5 (c 1, CHCl_3_); R_f_ 0.31 (dichloromethane/ethanol, 60:1, *v*/*v*); IR (KBr) ν_max_ 3567, 3310, 2963, 2124, 1739, 1457, 1262 cm^−1^; ^1^H NMR (600 MHz, CDCl_3_): δ 4.69 (1H, s, H-29), 4.59 (1H, s, H-29), 4.32 (1H, m, H-3), 4.25 (2H, t, *J* = 7.2 Hz, O*CH*_2_), 3.81 (1H, d, *J* = 10.8 Hz, H-28), 3.34 (1H, d, *J* = 10.8 Hz, H-28), 2.60 (2H, m, OCH_2_*CH*_2_), 2.40 (1H, m, H-19), 2.02 (1H, t, *J* = 2.4 Hz, C≡*CH*), 1.69 (3H, s, CH_3_), 1.08 (3H, s, CH_3_), 1.04 (3H, s, CH_3_), 0.99 (3H, s, CH_3_), 0.93 (3H, s, CH_3_), 0.86 (3H, s, CH_3_); ^13^C NMR (150 MHz, CDCl_3_): δ 155.0, 150.5, 109.7, 85.7, 79.5, 70.2, 65.0, 55.4, 50.3, 48.7, 47.8, 42.7, 40.9, 38.3, 38.1, 37.3, 37.0, 34.1, 34.0, 29.7, 29.1, 27.8, 27.0, 25.1, 23.6, 20.8, 19.1, 18.1, 16.4, 16.1, 15.9, 14.7; HRAPCIMS *m*/*z*: 537.3830 C_35_H_53_O_4_ (calcd. 537.3944). (Appendix A, ^1^H NMR and ^13^C NMR spectra of 3-modified betulin derivatives **5**–**14**, Appendix A).

### 3.6. General Procedure for the Synthesis of Betulinic Aldehyde Derivatives ***15**–**24***

To a solution of corresponding 3-substituted triterpene **5–14** (0.22 mmol) in anhydrous dichloromethane (3.5 mL) was added PCC (0.07 g, 0.33 mmol). The reaction mixture was stirred at room temperature for 2h. Next, the silica gel (1 g) was added into the mixture and stirred another 10 min. The reaction mixture was filtered off. The filtrate was concentrated under reduced pressure and obtained residue was purified by column chromatography (CH_2_Cl_2_-EtOH 60:1, *v*/*v*) to afford compounds **15**–**22** with 61–84% yields.

3-Propanoylbetulinic aldehyde (**15**) Yield 80%; mp 155–157 °C; [α]_D_^20^ +2.3 (c 1, CHCl_3_); R_f_ 0.65 (dichloromethane/ethanol, 60:1, *v*/*v*); IR (KBr) ν_max_ 2938, 1731, 1457, 1260 cm^−1^; ^1^H NMR (600 MHz, CDCl_3_): δ 9.70 (1H, s, CHO), 4.77 (1H, s, H-29), 4.65 (1H, s, H-29), 4.50 (1H, m, H-3), 2.88 (1H, m, H-19), 2.35 (2H, q, *J* = 7.2 Hz, *CH*_2_CH_3_), 1.68 (3H, s, CH_3_), 1.18 (3H, t, *J* = 7.2 Hz, CH_2_*CH*_3_), 1.00 (3H, s, CH_3_), 0.94 (3H, s, CH_3_), 0.87 (3H, s, CH_3_), 0.86 (3H, s, CH_3_), 0.85 (3H, s, CH_3_); ^13^C NMR (150 MHz, CDCl_3_): δ 206.7, 174.3, 149.7, 110.2, 80.6, 59.3, 55.4, 50.4, 48.1, 47.6, 42.6, 40.9, 38.7, 38.4, 37.9, 37.1, 34.3, 33.2, 29.9, 29.2, 28.8, 28.1, 28.0, 25.5, 23.7, 20.8, 19.0, 18.2, 16.5, 16.2, 15.9, 14.3, 9.4; HRAPCIMS *m*/*z*: 495.3486 C_33_H_51_O_3_ (calcd. 495.3438).

3-(2-Propenoyl)betulinic aldehyde (**16**) Yield 83%; mp 262–266 °C; [α]_D_^20^ +3.2 (c 1, CHCl_3_); R_f_ 0.71 (dichloromethane/ethanol, 60:1, *v*/*v*); IR (KBr) ν_max_ 2948, 1724, 1452, 1273 cm^−1^; ^1^H NMR (600 MHz, CDCl_3_): δ 9.69 (1H, s, CHO), 6.39 (1H, m, CH=*CH*_2_), 6.13 (1H, m, *CH*=CH_2_), 5.82 (1H, m, CH=*CH*_2_), 4.78 (1H, s, H-29), 4.65 (1H, s, H-29), 4.57 (1H, m, H-3), 2.89 (1H, m, H-19), 1.69 (3H, s, CH_3_), 1.00 (3H, s, CH_3_), 0.90 (3H, s, CH_3_), 0.88 (3H, s, CH_3_), 0.86 (3H, s, CH_3_), 0.85 (3H, s, CH_3_); ^13^C NMR (150 MHz, CDCl_3_): δ 206.7, 166.1, 149.7, 130.1, 129.2, 110.2, 81.1, 59.3, 55.4, 50.4, 48.0, 47.6, 42.6, 40.9, 38.7, 38.4, 38.0, 37.1, 34.3, 33.2, 29.9, 29.2, 28.8, 28.0, 25.5, 23.7, 20.8, 19.0, 18.2, 16.6, 16.2, 15.9, 14.3; HRAPCIMS *m*/*z*: 493.3326 C_33_H_49_O_3_ (calcd. 493.3681).

3-(3-Cyclopropyl-2-propynoyl)betulinic aldehyde (**17**) Yield 72%; mp 220–223 °C; [α]_D_^20^ +1.5 (c 1, CHCl_3_); R_f_ 0.76 (dichloromethane/ethanol, 60:1, *v*/*v*); IR (KBr) ν_max_ 2939, 2224, 1729, 1450, 1250 cm^−1^; ^1^H NMR (600 MHz, CDCl_3_): δ 9.69 (1H, s, CHO), 4.78 (1H, s, H-29), 4.65 (1H, s, H-29), 4.58 (1H, m, H-3), 2.88 (1H, m, H-19), 1.68 (3H, s, CH_3_), 1.40–1.36 (5H, m, CH, CH_2_), 0.95 (3H, s, CH_3_), 0.93 (3H, s, CH_3_), 0.89 (3H, s, CH_3_) 0.88 (3H, s, CH_3_), 0.86 (3H, s, CH_3_); ^13^C NMR (150 MHz, CDCl_3_): δ 207.2, 154.5, 150.2, 110.7, 93.2, 83.2, 69.4, 59.9, 56.0, 50.9, 48.6, 48.1, 43.1, 41.4, 39.2, 39.0, 38.5, 37.6, 34.8, 33.8, 30.4, 29.8, 29.3, 28.4, 26.0, 24.1, 21.3, 19.5, 18.7, 17.1, 16.7, 16.4, 14.8, 9.7, 1.6; HRAPCIMS *m*/*z*: 531.3479 C_36_H_51_O_3_ (calcd. 531.3838).

3-Phenylpropynoylbetulinic aldehyde (**18**) Yield 84%; mp 197–199 °C; [α]_D_^20^ +2.2 (c 1, CHCl_3_); R_f_ 0.73 (dichloromethane/ethanol, 60:1, *v*/*v*); IR (KBr) ν_max_ 2944, 2228, 1706, 1442, 1284 cm^−1^; ^1^H NMR (600 MHz, CDCl_3_): δ 9.67 (1H, s, CHO), 7.53–7.29 (5H, m, Ar-H), 4.69 (1H, s, H-29), 4.58 (1H, m, H-3), 4.56 (1H, s, H-29), 2.81 (1H, m, H-19), 1.67 (3H, s, CH_3_), 0.91 (3H, s, CH_3_), 0.88 (3H, s, CH_3_), 0.86 (3H, s, CH_3_), 0.85 (3H, s, CH_3_), 0.79 (3H, s, CH_3_); ^13^C NMR (150 MHz, CDCl_3_): δ 206.7, 154.2, 149.7, 132.9, 130.5, 128.5, 119.9, 110.2, 85.8, 83.2, 81.1, 59.4, 55.4, 53.4, 50.4, 48.0, 47.6, 42.6, 40.8, 38.7, 38.7, 38.5, 38.0, 37.1, 34.2, 33.2, 29.8, 29.2, 28.8, 27.9, 25.5, 23.6, 20.9, 20.8, 19.0, 18.2, 16.6, 16.2, 15.9, 14.2; HRAPCIMS *m*/*z*: 567.3474 C_39_H_51_O_3_ (calcd. 567.3838).

3-(2-Butynoyl)betulinic aldehyde (**19**) Yield 63%; mp 178–180 °C; [α]_D_^20^ +0.8 (c 1, CHCl_3_); R_f_ 0.69 (dichloromethane/ethanol, 60:1, *v*/*v*); IR (KBr) ν_max_ 2940, 2244, 1697, 1457, 1268 cm^−1^; ^1^H NMR (600 MHz, CDCl_3_): δ 9.60 (1H, s, CHO), 4.69 (1H, s, H-29), 4.56 (1H, s, H-29), 4.50 (1H, m, H-3), 2.80 (1H, m, H-19), 2.02 (3H, s, C≡C*CH*_3_), 1.69 (3H, s, CH_3_), 0.90 (3H, s, CH_3_), 0.84 (3H, s, CH_3_), 0.81 (3H, s, CH_3_), 0.80 (3H, s, CH_3_), 0.77 (3H, s, CH_3_); ^13^C NMR (150 MHz, CDCl_3_): δ 205.6, 152.9, 148.7, 109.2, 83.9, 81.8, 71.8, 58.3, 54.4, 49.3, 47.0, 46.5, 41.5, 39.8, 37.6, 37.4, 36.9, 36.0, 33.2, 32.2, 28.8, 28.7, 28.2, 26.9, 24.5, 22.6, 19.7, 18.0, 17.1, 15.5, 15.2, 14.9, 13.2, 2.8; HRAPCIMS *m*/*z*: 505.3684 C_34_H_49_O_3_ (calcd. 505.3681).

3-Ethoxycarbonylbetulinic aldehyde (**20**) Yield 70%; mp 155–158 °C; [α]_D_^20^ +4.1 (c 1, CHCl_3_); R_f_ 0.63 (dichloromethane/ethanol, 60:1, *v*/*v*); IR (KBr) ν_max_ 2942, 1736, 1465, 1253 cm^−1^; ^1^H NMR (600 MHz, CDCl_3_): δ 9.69 (1H, s, CHO), 4.78 (1H, s, H-29), 4.65 (1H, s, H-29), 4.33 (1H, m, H-3), 4.21 (2H, q, *J* = 7.2 Hz, O*CH*_2_), 2.88 (1H, m, H-19), 1.70 (3H, s, CH_3_), 1.36 (3H, t, *J* = 7.2 Hz, CH_2_*CH*_3_), 0.99 (3H, s, CH_3_), 0.98 (3H, s, CH_3_), 0.94 (3H, s, CH_3_), 0.87 (3H, s, CH_3_), 0.86 (s, CH_3_, 3H); ^13^C NMR (150 MHz, CDCl_3_): δ 206.7, 155.3, 149.7, 110.2, 85.1, 63.7, 59.3, 55.4, 50.4, 48.0, 47.6, 42.6, 40.8, 38.7, 38.4, 38.0, 37.1, 34.2, 33.2, 29.9, 29.2, 28.8, 27.9, 25.5, 23.7, 20.8, 19.0, 18.1, 16.4, 16.2, 15.9, 14.3, 14.2; HRAPCIMS *m*/*z*: 511.3429 C_33_H_51_O_4_ (calcd. 511.3787).

3-Propoxycarbonylbetulinic aldehyde (**21**) Yield 75%; mp 168–171 °C; [α]_D_^20^ +1.1 (c 1, CHCl_3_); R_f_ 0.69 (dichloromethane/ethanol, 60:1, *v*/*v*); IR (KBr) ν_max_ 2941, 1735, 1465, 1266 cm^−1^; ^1^H NMR (600 MHz, CDCl_3_): δ 9.69 (1H, s, CHO), 4.77 (1H, s, H-29), 4.65 (1H, s, H-29), 4.33 (1H, m, H-3), 4.11 (2H, t, *J* = 6.6 Hz, O*CH*_2_), 2.89 (1H, m, H-19), 1.74 (2H, m, *CH*_2_CH_3_), 1.70 (3H, s, CH_3_), 0.99 (3H, s, CH_3_), 0.97 (3H, t, *J* = 7.2 Hz, CH_2_*CH*_3_), 0.93 (3H, s, CH_3_), 0.92 (3H, s, CH_3_), 0.87 (s, CH_3_, 3H), 0.86 (s, CH_3_, 3H); ^13^C NMR (150 MHz, CDCl_3_): δ 205.7, 154.4, 148.7, 109.2, 84.1, 68.2, 58.3, 54.4, 49.3, 47.0, 46.5, 41.5, 39.8, 37.7, 37.0, 36.0, 33.2, 32.2, 28.8, 28.7, 28.2, 27.8, 26.8, 24.5, 22.7, 21.0, 19.7, 18.0, 15.4, 15.2, 13.2, 9.2; HRAPCIMS *m*/*z*: 525.3601 C_34_H_53_O_4_ (calcd. 525.3944).

3-Allyloxycarbonylbetulinic aldehyde (**22**) Yield 75%; mp 142–145 °C; [α]_D_^20^ +2.1 (c 1, CHCl_3_); R_f_ 0.73 (dichloromethane/ethanol, 60:1, *v*/*v*); IR (KBr) ν_max_ 2941, 1739, 1458, 1271 cm^−1^; ^1^H NMR (600 MHz, CDCl_3_): δ 9.60 (1H, s, CHO), 5.88 (1H, m, *CH*=CH_2_), 5.29 (1H, m, CH=*CH*_2_), 5.20 (1H, m, CH=*CH*_2_), 4.68 (1H, s, H-29), 4.56 (2H, m, O*CH*_2_), 4.55 (1H, s, H-29), 4.25 (1H, m, H-3), 2.80 (1H, m, H-19), 1.69 (3H, s, CH_3_), 0.91 (3H, s, CH_3_), 0.90 (3H, s, CH_3_), 0.78 (3H, s, CH_3_), 0.77 (3H, s, CH_3_), 0.71 (3H, s, CH_3_);^13^C NMR (150 MHz, CDCl_3_): δ 206.7, 155.1, 149.7, 131.9, 118.7, 110.2, 85.5, 68.2, 59.3, 55.4, 50.4, 48.0, 47.5, 42.6, 40.8, 38.7, 38.4, 38.0, 37.1, 34.2, 33.2, 29.8, 29.2, 28.8, 27.9, 25.5, 23.6, 20.8, 19.0, 18.1, 16.4, 16.2, 15.9, 14.3; HRAPCIMS *m*/*z*: 539.3763 C_34_H_51_O_4_ (calcd. 539.3787).

3-Propargyloxycarbonylbetulinic aldehyde (**23**) Yield 81%; mp 145–147 °C; [α]_D_^20^ +2.5 (c 1, CHCl_3_); R_f_ 0.72 (dichloromethane/ethanol, 60:1, *v*/*v*); IR (KBr) ν_max_ 3311, 2941, 2131, 1747, 1456, 1274 cm^−1^; ^1^H NMR (600 MHz, CDCl_3_): δ 9.67 (1H, s, CHO), 4.75 (1H, s, H-29), 4.71 (2H, d, *J* = 2.4 Hz, O*CH*_2_), 4.63 (1H, s, H-29), 4.33 (1H, m, H-3), 2.86 (1H, m, H-19), 2.51 (1H, t, *J*= 2.4 Hz, C≡*CH*), 1.69 (3H, s, CH_3_), 0.97 (3H, s, CH_3_), 0.92 (3H, s, CH_3_), 0.91 (3H, s, CH_3_), 0.85 (3H, s, CH_3_), 0.84 (3H, s, CH_3_); ^13^C NMR (150 MHz, CDCl_3_): δ 206.6, 154.6, 149.7, 110.2, 86.2, 77.2, 76.8, 75.4, 59.3, 55.4, 54.9, 50.3, 48.0, 47.5, 42.6, 40.8, 38.7, 38.3, 38.0, 37.0, 34.2, 33.2, 29.8, 29.2, 28.8, 27.8, 25.5, 23.6, 20.8, 19.0, 18.1, 16.3, 16.1, 15.9, 14.2; HRAPCIMS *m*/*z*: 521.3643 C_34_H_49_O_4_ (calcd. 521.3631).

3-(3-Butynyloxycarbonyl)betulinic aldehyde (**24**) Yield 61%; mp 143–145 °C; [α]_D_^20^ +3.3 (c 1, CHCl_3_); R_f_ 0.68 (dichloromethane/ethanol, 60:1, *v*/*v*); IR (KBr) ν_max_ 3284, 2940, 2358, 1746, 1456, 1274 cm^−1^; ^1^H NMR (600 MHz, CDCl_3_): δ 9.69 (1H, s, CHO), 4.77 (1H, s, H-29), 4.65 (1H, s, H-29), 4.33 (1H, m, H-3), 4.24 (2H, t, *J* = 7.2 Hz, O*CH*_2_), 2.88 (1H, m, H-19), 2.60 (2H, m, OCH_2_*CH*_2_), 2.02 (1H, t, *J* = 2.4 Hz, C≡*CH*), 1.71 (3H, s, CH_3_), 0.99 (3H, s, CH_3_), 0.93 (3H, s, CH_3_), 0.86 (3H, s, CH_3_), 0.80 (3H, s, CH_3_), 0.77 (3H, s, CH_3_); ^13^C NMR (150 MHz, CDCl_3_): δ 206.7, 155.0, 149.7, 110.2, 85.7, 70.2, 65.0, 59.3, 47.5, 42.6, 40.8, 38.1, 37.1, 33.2, 29.8, 29.2, 28.8, 27.8, 25.5, 23.6, 20.8, 19.1, 19.0, 18.1, 16.4, 16.2, 15.9, 14.2; EIMS *m*/*z* 536 [M]^+^ (14), 189 (100). (Appendix A, ^1^H NMR and ^13^C NMR spectra of 3-modified betulinic aldehyde derivatives **15**–**24**, Appendix A).

### 3.7. Antiproliferative Activity—MTT and SRB Assays

#### 3.7.1. Human Cell Lines

The applied human cell lines such as MV-4-11 (biphenotypic B myelomonocytic leukaemia), A549 (lung adenocarcinoma), Du-145 (prostate), Hs294T (melanoma), MCF-7 (breast adenocarcinoma) and MCF-10A (normal mammary gland) were obtained from the American Type Culture Collection (ATCC, Rockville, MD, USA). The all cell lines are maintained at the Cell Culture Collection of the Institute of Immunology and Experimental Therapy (Wrocław, Poland). The cells were plated in 96-well plates (Sarstedt, Newton, MA, USA) at a density of 1 × 10^4^ cells per well in 100 µL of appropriate culture medium overnight.

Biphenotypic B myelomonocytic leukaemia (MV-4-11) cells were cultured in RPMI 1640 medium (Gibco, Scotland, UK) with GlutaMAX (Thermo-Fisher Scientific, Warsaw, Poland) adjusted to contain 1.0 mM sodium pyruvate (Sigma-Aldrich, Chemie GmbH, Steinheim, Germany). Human lung adenocarcinoma (A549) cells were cultured in mixture RPMI 1640 and Opti-MEM medium (Gibco, Scotland, UK) supplemented with GlutaMAX (ThermoFisher Scientific, Warsaw, Poland) and 2mM L-glutamine adjusted to contain 1.0 mM sodium pyruvate. Human prostate (Du-145) and human breast adenocarcinoma (MCF-7) cells were cultured in Eagle medium (IIET, Wrocław, Poland), supplemented with 2 mM L-glutamine adjusted to contain 1.0 mM sodium pyruvate. Human melanoma (Hs294T) cells were cultured in Dulbecco medium (Gibco, Scotland, UK) supplemented with 2 mM L-glutamine. The normal human mammary gland cells (MCF-10A) were cultured in Ham’s F-12 medium containing 2 mM L-glutamine, 5% horse serum, epidermal growth factor EGF (100 μL/100 mL of medium), hydrocortisone (1 mL/ 100 mL of medium), insulin (100 μL/100 mL of medium) and cholera toxin (0.1 mL/100 mL of medium). All applied culture media except MCF-10A were supplemented with 10% of FBS (MV-4-11 cultured with FBS derived from Sigma-Aldrich, Chemie GmbH, Steinheim, Germany, the other cell lines FBS HyClone Thermo-Fisher Scientific, Warsaw, Poland) and streptomycin (100 µg/mL), penicillin (100 U/mL) (both from Polfa, Tarchomin, Poland). The tested cell lines were grown at 37 °C in humid atmosphere saturated with 5% CO_2_.

#### 3.7.2. Antiproliferative Assays

In our studies were used two assays for determination the cytotoxic activity in vitro of triterpene derivatives. By dissolving 1 mg of each tested compound in 100 µL of DMSO, stock solutions at a 10 mg/mL were obtained. Next, the stock solutions were diluted in appropriate culture medium to reach the required concentrations of compounds ranging from 0.1 to 100 µg/mL. The MTT assay was used for the cytotoxicity screening against biphenotypic B myelomonocytic leukaemia (MV-4-11). The SRB assay was performed for adherent cells such as A549, Du-145, Hs294T, MCF-7 and MCF-10A. Both assays were carried as described by Wietrzyk et al. [38]. The triterpenes in given concentration were examined in triplicates in each experiment, which was repeated 3 times. The results of anticancer activity in vitro were presented as an IC_50_ in µM. Cisplatin and betulin **1** were used as reference substances. 

### 3.8. In Silico Study

The physicochemical parameters of compounds **5**–**24** such as penetration drugs by BBB (log BB), lipophilicity (cLogP), topological polar surface area (tPSA), hydrogen bond donors (HBD) and hydrogen bond acceptors (HBA) topological polar surface area (tPSA) were calculated using the ACD/Labs software [39].

The target macromolecule for molecular docking studies were used from the Protein Data Bank (available online: https://www.rcsb.org/). In our studies we applied 3D crystal structure of Akt1 (ID: 3QKK).

The three-dimensional (3D) structures of tested triterpenes required were generated in their low-energy conformation using Gaussian 16 (revision A.03) computer code [40] at the density functional theory (DFT, B3LYP [41]) and 6–311 + G(d,p) basis sets. In docking study was used Genetic Optimisation for Ligand Docking (GOLD) 5.6.3 [42] The receptors for docking were prepared using a Gold packet of the Hermes visualizer. The docking was defined for all the protein residues within 10 Å of the reference ligands that accompanied the downloaded protein complexes. For all other parameters a default values were used and the complexes were submitted to 10 genetic algorithm runs using the GOLDScore fitness function. The BIOVIA Discovery Studio virtual environment was used to visualize molecular docking details [43].

## 4. Conclusions

The synthesis, biological activity and in silico studies of newly 3-substituted derivatives of betulin and betulinic aldehyde were presented. The molecular structures of triterpenes **5**–**24** were confirmed by ^1^H NMR, ^13^C NMR, IR and HR-MS spectra. Antiproliferative activity of compounds was evaluated in vitro against five human cancer cell lines such as biphenotypic B myelomonocytic leukaemia (MV-4-11), adenocarcinoma (A549), prostate (Du-145), melanoma (Hs294T), breast adenocarcinoma (MCF-7) and normal human mammary gland (MCF-10A). The compound 9 having 2-butynoyl group, showed the highest anticancer activity against three tested cancer cell lines: Du-145, Hs294T and MCF-7. Molecular docking studies were performed to investigate the binding mechanism of the 3-substituted derivatives of betulin and betulinic aldehyde with Akt protein. Our studies showed significant difference in location of triterpene 9 in Akt CAT domain when compared to betulin 1. Moreover, the hydrophobic behaviour of compound 9 may be explanation of its potent inhibitor activity.

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
