# Peer review of "Biological Activity and In Silico Study of 3-Modified Derivatives of Betulin and Betulinic Aldehyde"

_ijms, 2019, doi:10.3390/ijms20061372_

Round 1
Reviewer 1 Report
The authors illustrate the preparation of ester derivatives of betulin and betulinic aldehyde, and investigate the potential anticancer activity of the synthetized compounds through in vitro assay. Moreover, some molecular docking has been performed to investigate the target of the most active compound.
The description of the synthesis is quite short: the authors should explain why they used THP as protecting group for the primary alcohol, as they could selectively oxidize it to aldehyde, avoiding the useless protection/deprotection steps. Moreover, PCC can be replace with more green and safety reagents in the oxidation step. Did the authors try other oxidation protocols?
In my opinion the authors should explicit clearly why they decide to synthetize these derivatives. Which kind of property they would like to reach with the substituents in 3-position? more potent compounds? better pharmacokinetic properties? more stability?
The final products are characterized, but several data are missed such as the peak assignment of the NMR (That’s could be important to avoid mistakes). For these reasons, I recomand the current report for publication after a revision of the manuscript:
The following comments have to be considered:
1) The authors have to explicit in the abstract the acronym ADME or TPSA
2) The authors have to provide higher figure resolution
3) I suggest to carefully checked the NMR assignment, several errors are present.

Author Response
Thank you for reviewing our manuscript. The revision of the paper is prepared according to all of your comments and highlighted with yellow.
Ans 1 We corrected description of the synthesis (line 102-105). We added in Scheme I the chemical formula of intermediates 5A-14A. Regarding your comment on our decision to use protection group instead of direct oxidation, I’d like to explain that in fact we previously tried it, but direct oxidation led to a complex mixture of products which was difficult to separate. Therefore a method with protection group worked better in our case. Thank you for your comment about PCC. In our next studies we will consider more green and safe reagents.
Ans 2 We decided to synthetize 3-modified derivatives to check their anticancer activity in vitro, as we expected to find compounds more potent than betulin (line 85-87).
Ans 3 In the abstract, we added the acronym ADME and TPSA (line 20-22).
Ans 4 We provided a higher resolution of the figures. According to the suggestion Figure 1 was changed.
Ans 5 We carefully checked the NMR assignment and corrected all mistakes. In the mass spectra of all compounds there are signals based on ions [M−H]-.
Best regards,
Authors
Reviewer 2 Report
The authors present the synthesis of betulin derivatives which have been assessed against a number of cancer cell lines. All compounds appear to be well characterised and a highlight of the manuscript is the improved inhibition of some compounds versus betulin. Although it does appear that selectivity versus healthy cells is compromised and this may be as a result of the introduction of reactive electrophiles in the structures (e.g. enoates). The electrophilic features should be featured in the discussion of SAR.
In silico data is also presented for oral drug likeness, target identification and compound binding. I feel that the conclusions drawn from these additions do not form a cohesive story for these compounds. For example the calculated physicochemical properties are considered individually rather than all together; TPSA alone is not a conclusive indication of BBB penetration. Additionally, no cell lines looked at were for neurological cancers.
As these compounds are very different to the standard kinase pharmacophore, more evidence should be provided to confirm the putative target.
These changes should be address before being re-considered for acceptance.
Author Response
Thank you for reviewing our manuscript. The revision of the paper is prepared according to all of your comments and highlighted with yellow.
Ans 1According to your suggestion, we changed the discussion:
A comparison of Selectivity Index (SI) for compounds 6, 9 and 16 show that triterpene 9 exhibits the highest SI values to Du-145, Hs294T and MCF-7 cell lines. Derivative 9, in contrast to compounds 6 and 16, does not have the α,β-unsaturated carbonyl group at the C-3 position. The activity of triterpenes 6 and 16 towards cancer as well normal cells may be due to the presence in their structure of a carbonyl group adjacent to the double bond. This atom group exhibits electron-accepting (electrophilic) properties and can react with electron-donor (nucleophilic) atoms present in living systems, for example, sulphydryl (-SH) group. (line 152-158).
Ans 2 We divided Subsection 2.3 into Subsubsection 2.3.1. and 2.3.2. First we described the molecular docking study and then the ADME parameters. Regarding your comment about lack of cohesive story between in silico and in vitro studies, I would like to explain that we decided to publish all calculated physicochemical properties of these compounds although we had no opportunity to test them towards neurological cancers. We made this decision hoping that other researchers might find this information useful.
Ans 3According to your suggestion, we added information about acetyl derivative of betulin inhibit the PI3K/Akt signaling pathway:
The anticancer activity of triterpenes towards tumor cells is related to their interaction with various molecular targets [18]. Zhuo et al. obtained a new acetyl derivative of betulin which effectively reduced the viability of cancer cells (Huh7). Described results of in vitro study demonstrated that this triterpene inhibited the PI3K/Akt signaling pathway by decreasing the expression levels of Akt [19]. (line 80-84).
Best regards,
Authors
Round 2
Reviewer 2 Report
The authors have addressed concerns and provide more background to help the cohesiveness of the manuscript. Justification for the presentation of calculated properties is also provide. I recommend this article is accepted for publication in the International Journal of Molecular Sciences.